# Thirst or dry mouth in dying patients?—A qualitative study of palliative care physicians' experiences

Maria Friedrichsen[1,2]*, Caroline Lythell[1,2], Tiny Jaarsma[2], Pier Jaarsma[3], Helene Ångström[1], Micha Milovanovic[2,4], Marit Karlsson[2], Anna Milberg[1,2], Hans Thulesius[5,6], Christel Hedman[7,8], Nana Waldréus[9,10], Anne Söderlund Schaller[2,11]

1 Palliative Education and Research Centre, Vrinnevi Hospital, Norrköping, Sweden, 2 Department of Health, Medicine and Caring Sciences, Linköping University, Linköping, Sweden, 3 Division of Society and Health, Department of Health, Medicine and Caring Sciences, Linköping University, Linköping, Sweden, 4 Department of Internal Medicine, Vrinnevi Hospital, Norrköping, Sweden, 5 Department of Medicine and Optometry, Faculty of Health and Life Sciences, Linnaeus University, Kalmar, Sweden, 6 Region Kronoberg, Växjö, Sweden, 7 Department of Molecular Medicine and Surgery, Karolinska Institutet, Stockholm, Sweden, 8 R & D Department, Stockholms Sjukhem Foundation, Stockholm, Sweden, 9 Department of Neurobiology, Care Sciences and Society, Division of Nursing, Karolinska Institutet, Huddinge, Sweden, 10 Theme Inflammation and Aging, Karolinska University Hospital, Huddinge, Sweden, 11 Pain and Rehabilitation Centre, Linköping, Sweden

* maria.friedrichsen@regionostergotland.se

## Abstract

### Introduction

Thirst and dry mouth are common symptoms among patients at the end of life. In palliative care today, there is a focus on mouth care to alleviate thirst. There are no qualitative studies on thirst from a physician's experience, which is why this study is needed.

### Purpose

This study aimed to explore palliative care physicians' experiences and views of thirst in patients at the end of life.

### Methods

A qualitative interview study with an inductive approach was carried out. Sixteen physicians working in specialised palliative care units in Sweden were included. The interviews were analysed with a reflexive thematic analysis.

### Results

The analysis resulted in three basic assumptions regarding thirst: It is dry mouth, not thirst; patients are dry in their mouth and thirsty; and, I do not know if they are thirsty. Further, four different themes regarding how to relieve thirst appeared: drips will not help thirst but cause harm; the body takes care of thirst itself; drips might help thirst; and, mouth care to relieve thirst or dry mouth.

**Data Availability Statement:** As the data for this study is qualitative (interview transcripts), it is sensitive, potentially identifying, and not able to be publicly shared. Requests to access these data can

be sent to The Swedish Ethical Review Authority (registrator@etikprovning.se) quoting the study approval number (Ref. 2019–04347).

**Funding:** This study was supported by a grant from The Sjöberg Foundation Fund, Sweden (number 20210114:6). The funders had no role in study design, data collection and analysis, decision to publish, or preparation of the manuscript.

## Conclusions

The palliative care physicians had different experiences regarding thirst, from thirst never arising, to a lack of awareness. They thought good mouth care worked well to alleviate the feeling of thirst and dry mouth. Most physicians did not want to give patients drips, while some did. This study indicates that there are many unanswered questions when it comes to thirst at end-of-life and that further research is needed.

## 1. Introduction

Thirst is the subjective sensation of a desire to drink water that cannot be ignored [1]. When severely ill patients approach the end of life they might stop eating and drinking (voluntarily or involuntarily) and among family physicians and palliative care (PC) physicians, this is seen as something natural [2, 3]. After patients stop eating and drinking, family physicians predict a median time of 7 days until death, and the most common symptoms before death were pain, fatigue, impaired cognitive functioning, and thirst or a dry throat [4].

The evidence regarding thirst in dying patients are inconsistent. Most previous studies regarding thirst are between 20–30 years old. Burge [5] reported that more than 50% of 52 terminally ill patients had a thirst intensity of 50 mm and pleasure from drinking of 61 mm on a VAS scale (100 mm, which suggests that patients suffer from thirst. Ellershaw's study [6] (n = 23) showed that 83% of patients who were very close to death were thirsty. Morita's study [7] (n = 88) showed that 44% of terminally ill patients had middle- severe thirst and 18% severe thirst. During the dying phase, patients drink on average 250 ml, but this varies significantly; 25–1650 ml [8].

Today, thirst studies are rare, as symptom scales used in PC do not usually measure thirst; dryness of mouth is measured instead. However, one study used a modified version of MSAS (Memorial Symptom Assessment Scale) where thirst was added to the original scale and this study found that thirst and existential distress were significantly correlated in terminally ill patients [9]. One solution to quench thirst is to provide artificial hydration. However, this is a controversial question. In one survey, it was found that PC physicians did not believe that artificial hydration would help symptoms such as thirst [10] but rather increase upper respiratory tract secretions (85%), ascites (73%), physical discomfort (72%) and dyspnoea (62%) [11]. As the evidence regarding thirst at end of life care shows, different results, it is interesting to study PC physicians experience regarding this phenomenon, as they meet terminally ill patients daily. Today there is limited knowledge regarding PC physicians' experience and reasoning regarding thirst. This study is a part of a larger project, The "Thirst Project", which studies thirst and ethical issues (ethical issues reported elsewhere) regarding thirst at the end of life from different perspectives, such as from dying patients, family members and health care personnel and biomedical. The purpose of the current study will qualitatively focus on PC physicians' experiences and views around thirst at the end of life for dying patients.

## 2. Materials and methods

### 2.1 Design

A qualitative, reflexive thematic design with an inductive analysis according to Braun & Clarke [12, 13] was used.

## 2.2 Sampling and setting

Data were collected in four purposively selected sized cities in Sweden with populations of between 44,000 to 1,000,000 during 2019–2022. All cities had advanced palliative care units, with physicians specialised in palliative medicine. One palliative care unit was chosen in each city. Purposeful sampling was used trying to achieve participants from different geographic locations, of different genders and ages (Table 1). The inclusion criteria were as follows: working as a physician in specialised PC or geriatric care; having at least five years' experience in working with dying patients; speaking Swedish. The head of the department or a senior physician in charge asked the physicians to participate. Before the interviews started, all physicians received information about the study by e-mail by their medical director or senior physician. Ethics committee approval was obtained from The Swedish Ethical Review Authority (Ref. 2019–04347). The study was conducted in accordance with the terms of the Helsinki Declaration, and written informed consent was obtained from each participant.

## 2.3 Data collection

Data were collected in 2019–2022 using face-to-face recorded interviews in 2019–2020 (n = 10) and telephone interviews in 2021–2022 (n = 6). The interviews lasted between 16 to 35 minutes, and a medical student (HÅ) with a special interest in PC and previous experience

**Table 1. Demographic data of the participating physicians.**

| *Gender* | |
|---|---|
| Male/female | 3/13 |
| *Age* | |
| M | 52 |
| Min-max | 37–66 |
| *Medical degree* | |
| Residents | - |
| Specialists | 3 |
| Consultants | 13 |
| *Specialised in** | |
| Palliative medicine | 10 |
| Geriatrics | 6 |
| Internal medicine | 4 |
| Oncology | 4 |
| General medicine | 1 |
| Gynaecology and obstetrics | 1 |
| Haematology | 1 |
| *Marital status* | |
| Married | 13 |
| Single | 3 |
| *Geographic location* | |
| Palliative care unit 1 | 6 |
| Palliative care unit 2 | 3 |
| Palliative care unit 3 | 1 |
| Palliative care unit 4 | 6 |

*More than one alternative can be chosen

**Table 2. Interview guide.**

1. Have you thought about whether end-of-life patients can be thirsty? If so, how?
2. Do you think end-of-life patients suffer from thirst? Justify!
3. Do you have a policy in your unit regarding thirst? Why/why not?
4. What are your colleagues' views on thirst in patients at the end of life?
5. Can there be any ethical problems with patients who are thirsty?
6. Do you check whether a patient at the end of life is thirsty? How do you assess whether a patient is thirsty? If yes, what checks do you make? If not, why not?
7. Do you do anything to quench the patient's thirst? What do you do? What do you think works best/worst?
8. Have you ever discussed thirst with relatives of patients at the end of life? Tell me!
9. How do you think one could work to quench thirst?
10. Is there anything I have not asked about thirst that you think is important to share?

of working as a nurse with dying patients conducted ten physical interviews. These interviews were carried out in a secluded room at the hospital in each town. The medical student transcribed these interviews for educational purposes. Due to the COVID-19 pandemic, six interviews were carried out by telephone. A research nurse with an MSc and experience of PC and interviewing conducted three interviews, and a nurse with a PhD and long experience of working with cancer patients and interviewing conducted three. A professional transcriber transcribed these six interviews verbatim. The semi-structured interview guide was developed by the research team, and addressed physicians' experiences and views of thirst and ethical challenges (ethical challenges not reported in this study) in patients at the end of life (Table 2). A pilot interview was conducted to test the understanding of the questions and the flow of the questions. No changes were made after the pilot interview. Participants were asked to talk freely, and follow-up questions were occasionally asked in order to achieve greater clarity, e.g., "Please, motivate; Please, tell me more; and Why/Why not?; and How?"

## 2.4 Data analysis

A reflexive thematic analysis according to Braun & Clarke [12, 13] was used. The design identified, analysed and interpreted patterns of meaning from the qualitative data, and was used to report concepts and assumptions underpinning the data, presented in themes.

A six-step process guided the analysis [14]. The analysis was recursive, requiring the researcher to move back and forth through the phases [12] and started with a familiarisation with the data, generating initial codes of relevance to the research question. Further, a generation of distinctive themes or sub-themes was created and a thematic map of the initial candidate themes was built to illustrate the relationships. To continue, each potential theme were scrutinised. Then we defined and named the themes and each theme and sub-theme was to be expressed in relation to both the dataset and the research question. Multiple extracts were used from the data items that inform a theme in order to convey the diversity of expressions of meaning across these data items.

Two authors (HÅ, MF) independently read all 10 transcripts that were conducted before the covid-19 pandemic and conducted coding and development of themes and sub-themes. The first author was an assistant professor with a PhD in palliative medicine with long experience of PC research and thematic analysis and the second was a medical student. The process of using different coders was to sense-check the ideas and explore alternative interpretations of the data. In 2021–2022, when the remaining interviews were conducted, the analysis continued. No new data emerged after 16 interviews. The final analysis was discussed in the research group until a consensus was reached. In order to further strengthen the credibility of the results, findings were demonstrated using quotes from participants.

## 3. Results

### 3.1 Assumptions regarding thirst

All the physicians had thought about thirst. In their clinical work, they did not prioritise thirst and seldom discussed it during rounds. There was no policy regarding thirst, but there was one for dryness of mouth and mouth care. Physicians described both conscious and unconscious patients in the terminal phase. Many were ambiguous regarding thirst, i.e. they presented facts regarding a possible clinical assessment regarding thirst at the same time as they said that they did not assess thirst. For an overview of the themes, see Table 3.

**3.1.1 It is dry mouth, not thirst.** When the physicians were asked about thirst, most of them replied that patients had dryness of the mouth and were not thirsty. They argued that patients took many drugs that induced dryness of mouth, and they ate and drank less, slept with an open mouth, and may have candida or other oral infections. Some mentioned research as a base for their standpoint, others that this was what they had learned when they started in the palliative care unit.

*I think it is often misinterpreted as thirst when it is dryness of the mouth.*

*I:9*

*I: How do you assess thirst?*

*R: No, not thirst. However, I assume that they have dryness in their mouth.*

*I:7*

*I think that we have a tendency to neglect thirst. Among all other symptoms, it is not prioritised.*

*I:4*

**3.1.2 Patients are dry in their mouths and thirsty.** Some physicians were clear that patients could be both thirsty and have dryness in their mouths. However, they had different standpoints for this. Some used detailed mouth care to assess thirst, while others believed that if patients feel thirst, it depends on their vital functions and consciousness at the end of life. Others argued that thirst is a drive or an instinct that humans have, and should be considered.

*I think that one should ask for it (thirst). You should think that it is painful to be thirsty. I do not think that is controversial, as all of us that are not dying, have something to drink when we are thirsty and that is a basic instinct.*

*I:10*

**Table 3. Overview of the themes.**

| Assumptions regarding thirst | How to relieve thirst |
| --- | --- |
| It is dry mouth, not thirst | Drips might help thirst |
| Patients are dry in their mouth and thirsty | The body take care of thirst itself |
| I do not know if they are thirsty | Drips will not help thirst but cause harm |
| | Mouth care to relieve thirst or dry mouth |

*If we dab the patient in the mouth, especially between the lips and the teeth, then you can see whether they are satisfied.*

*I:*16

Some physicians distinguished between dryness of the mouth and thirst. They argued that there was a difference between thirst and dryness of the mouth, where the first is a central experience caused by dehydration and that dryness of the mouth is secondary to thirst.

*Thirst is dehydration, dryness of the mouth is dryness in the mucous membranes, and that is what you can get when you are thirsty. As I interpret thirst, it is a central experience in our brain, which is how I think. Dryness of the mouth is more of a symptom, one of the symptoms of thirst.*

*I:*13

Others meant that dryness of the mouth produces many more symptoms than thirst as it was easier to assess. A patient might be thirsty but it is not possible to see nor measure it.

*Dryness of the mouth and thirst occur together, but it is more common that dryness of the mouth makes a person thirsty rather than the opposite. That is how I think, but thirst does not produce that many symptoms.*

*I:*6

Others were clear that thirst was something that patients experience locally in their mouth as humans have thirst receptors there and it should therefore be relieved with mouth care.

*My feeling is that thirst is directly in the mouth, so it is extremely important to moisturise and keep the mouth clean.*

*I:*16

*The mouth cavity has thirst receptors, so if you keep the mouth moisturised... then it is a very good way to relieve thirst.*

*I:*11

**3.1.3 I do not know if they are thirsty.** Some physicians were honest and said that they did not know whether patients were thirsty or not, especially if patients were unconscious, they just believed or speculated around it. They relied on the other team members' competence and skills in these matters, such as nurses and assistant nurses.

*I*: *Do you believe that dying patients suffer from thirst*?

*R*: *Well, what I think is... they suffer when they are very dry. Well, when they are dehydrated. Then it is hard to say if they are thirsty. I don't know.// No, I don't know that, no. However, I can assume that, but that is an assumption... that they show that they are thirsty through body language, that they want something.*

*I:*3

*No, I do not check if a patient at the end of life is thirsty. If the patient is, awake and conscious then I can ask him/her, among other questions. If they are thirsty, they will tell me. I am not looking in the mouth for thirst, but rather the condition of the mouth, dryness of mouth, candida, or anything else that can be unpleasant for the patient.*

*I: Do you think that good mouth care relieves thirst?*

*R: Yes, but I do not know. I cannot know that. I do not even know to what extent they experience thirst.*

*I:8*

### 3.2 How to relieve thirst

**3.2.1 Drips will not help thirst but cause harm.**   The PC physicians experienced that infusions are in most cases of no benefit for a dying patient. They argue that the body cannot use the prescribed fluids given, as it starts to prepare to die. Instead, the fluids will cause lung oedema and other oedemas that will cause more suffering.

*The indication for infusion might be to relieve thirst, but there is a risk if you have a patient with heart failure, that the patient will drown as a result of lung oedema. We know that drips at the end of life tend to end up in the wrong place, they lead to oedema.*

*I:10*

Some physicians said that there is no evidence that infusions will relieve the suffering of thirst and that they should therefore not give it. A requirement for an intravenous entrance was in itself a big issue at the final phase of life.

*I do not think that drips relieve thirst, it puts strain on the organs, and I am convinced that you will get rattles. . .//Drips at the end of life, require intravenous access and then it becomes a hospital matter.*

*I:7*

**3.2.2 The body takes care of thirst itself.**   Some physicians claimed that the human body prepares itself for dying, by shutting down organs and adapting to the process of dying, therefore the patient does not feel thirst, as the body takes care of the available resources.

*When the body enters this last phase of life, it then starts breaking down all the remaining reserves and muscles. Fluids are then released that are bound in both fat and muscles. It has been reported in studies that it is not a kind of osmotic dehydration. . . as the body regulates this itself. The body is fantastic, as it knows when it is going to die. . . how it should adapt.*

*I:14*

Others gave examples that there were certain patients with for example ascites, that even though they were dying, they still produced urine, and they claimed that the body took care of fluids itself and used it.

*Some humans are like camels and dromedaries; they have a store of fluids, like ascites or oedema. Maybe it is catabolic, these last days in life. You get surprised that they can still produce 0.5 l of urine, without drinking anything during their last week. Where does it come from*?

I:7.

**3.2.3 Drips might help thirst.** A few physicians claimed that there were certain cases when patients would benefit from infusions, even though it was unusual, as patients might be "real" thirsty or have other symptoms relieved by it, and then there is an indication for infusions. The physicians emphasise that the infusion has to be carefully evaluated to ensure that it does not cause any harm.

*I am positive to drips if there is a clear indication, as if there are signs of thirst and dryness of mouth, and there are other clinical signs as well.*

I:5

*I have a patient right now that has a drip, even though her life is very limited. She cannot eat or drink and has problems swallowing. They removed the drip at the hospital, but once she comes to us, we put her on a drip. She thinks that it is very good and nice.*

I:13.

**3.2.4 Mouth care to relieve thirst or dry mouth.** Physicians' that assess thirst do it in the same as way when they look for dehydration. Others have their own experience of how to assess whether a patient seems to be thirsty or not.

*You notice how they react to mouth care. They may suck on the liquid on the oral foam swab, and then you understand that they are thirsty, so you have to give them more, or they pinch their mouth together which indicates that they do not want any fluid. Then you have to respect that.*

I:14

If thirst nevertheless started to develop, it started days or weeks before the patient's death, but it seldom starts during the last 24 hours before death. Thirst could easily be relieved via oral care by nursing staff giving the patient small sips of water from time to time or by moistening the patient's mouth with water-filled oral foam swabs. A few physicians knew that the oral mucosa could absorb water, so mouth care could relieve thirst. Other physicians mentioned that it was the thirst receptors that quenched the thirst when water was given.

*If a patient is given mouth care regularly, so you are trying to moisturise the mouth, then they resorb a small part in the mucous membrane*:

I:1

*The mild thirst can usually be relieved by just moisturising the mouth, as you take up fluids via the oral mucosa as well.*

I:5

Some physicians said that the water used should be at room temperature while others said that it should be cold. Some suggested that mouth care should be given every 10 minutes while others suggested every 4 hours.

*We have guidelines that say that every 20 minutes we should moisturise the patient's mouth cavity so that he/she will not be dry or thirsty.*

*I*:14

*We have routines every 4 hours, but I cannot say this for sure. I think nurses and assistant nurses are better at answering that question, as I think they have a feeling for what is needed.*

*I*:15

Most physicians did not make a difference between the relieving interventions for dryness of mouth or thirst. They suggested ice cubes, oil, mouth spray and saliva stimulants. A few physicians mentioned that patients with heart failure should take their ACE- inhibitors for as long as possible as this gave relief from thirst.

## 4. Discussion

This unique study contributes with knowledge, that PC physicians had various experiences regarding thirst for patients at the end of life. Many do not consider thirst as a major problem, i.e., dry mouth is considered a greater problem. Some study participants mix up dry mouth with thirst. There are also different experiences of how mouth care should be carried out. As for thirst relief, some PC physicians never gave their patients artificial infusions while others did so when indicated.

In the current study, PC physicians raise dry mouth as a bigger problem than thirst, which is in line with Espen guidelines [15], that mention that fluids are not always needed at the end of life, as patients *"frequently experience dryness of the mouth. . . but rarely hunger and thirst"*. Espen guidelines [15] are building this hypothesis on the basis that thirst results from an unpleasant dryness of the oral cavity. However, other PC physicians in the current study perceived that thirst were present in dying patients and associated thirsty patients with the level of consciousness, dehydration, or thirst receptors in the mouth, which should be considered in clinical practise. In a study among terminally ill cancer patients [7], they found significant correlations between higher levels of thirst and dehydration defined by ANP level (atrial natriuretic peptide) ($<$ or $=$ 15 pg/ml), hyper- osmolality ($>$ or $=$ 300 mosmol/kg), oral intake, survival, performance status, gastrointestinal cancer, vomiting, and stomatitis. This is clinically important to keep in mind when meeting terminally ill patients. Some physicians perceived that the solution to relieve thirst was located in the mouth, which is supported by these preclinical studies. Assessing thirst is very important, as physicians should be able to determine whether a patient requires more than mouth care.

All of the participants felt that mouth care was important to relieve both thirst and dry mouth, but there were different views as to how it should be done. Should the water be cold or should it be at room temperature for patients at the end of life? There is no clear answer to this question, as it has not been studied in this context. With regard to quenching thirst in healthy people, it is a known fact that cold water is more appreciated than water that is at room temperature [16, 17]. This question should be studied further. There were also different experiences as to how often mouth care should be given, with a wide range from every 10 minutes to every 4 hours. This seems to be very different when studying guidelines for mouth care. The guidelines recommends that dying patients' level of thirst should be checked and that they

should be given frequent sips of fluids [18], mouth care as often as a patient tolerates it [19], or offered at least every 30 minutes [20]. In one study on thirsty healthy humans, the participants' gargled water and a thirst reduction sustained for 15 minutes, and longed for 30 minutes. Another study in long-term facilities showed that thirst could be alleviated with small amounts of fluids, and by application of ice chips; thirst relief were longing for one to several hours [21].

With regard to relief from thirst with the use of an artificial infusion, the PC physicians had different experiences; some never gave that to terminally ill patients while others did if they had indications for it. This standpoint was criticised by Ivanović et al., arguing that in PC there is an assumption that the symptom "thirst" is rather a sensation of dryness in the mouth than the need to administer hydration [22]. The question of artificial infusion has been discussed over the years. In the current study, many of the physicians claimed that artificial infusion would harm the patient resulting in lung oedema. These experiences are in line with a recent study among PC physicians where artificial infusions were thought to worsen oedema and respiratory symptoms [10]. On the contrary, a review study [23] regarding artificial infusion reported the opposite, where six out of eight studies did not report higher respiratory secretions, three studies of four did not report dyspnoea, and three older studies did not report any impact on thirst/dry mouth. Nevertheless, another question is whether artificial infusions will alleviate thirst.

The evidence is inconsistent as to whether artificial hydration should relieve thirst or not. Despite IV hydration regimens from 500 ml to 3000 ml among 19 dying patients, six experienced mild thirst, eight moderate thirst, and four severe thirst [24]. Even though 70% of the patients had signs of fluid retention, there was little correlation between these signs and the amount of fluids received. Other studies reported that subcutaneous infusions of max 1000 ml/24 h had a significant effect on nausea and thirst, but not on thirst after 48 hours [25, 26]. Recent studies have shown that IV infusion did not prolong survival nor significantly improved the dehydration symptoms, but improved the quality of dying [27]. Other studies did not find any difference when administering IV artificial infusions to improve delirium, quality of life, symptom burden, consciousness and survival [28–30]. A review study concluded that artificial hydration is of a more symbolic value for the family rather than a help for dehydration, thirst or dry mouth [31] as well as there are cultural beliefs regarding hydration.

To conclude, the PC physicians perceived dry mouth as a bigger problem than thirst. Physicians had different experiences regarding thirst, from thirst never arising, to a lack of awareness and an awareness. They thought good mouth care worked well to alleviate the feeling of both thirst and dry mouth. Thirst seems to be a forgotten sign among PC physicians, and has to be more emphasised in both research and in clinical practise. Future research should focus on the prevalence of thirst in a larger population, as well as patients' and family members' experience of thirst as well as other health care professionals' experiences.

## 4.1 Strengths and limitations

There are a number of methodological limitations in our study. First, this study is based on information gathered from a limited number of physicians in Sweden. However, no new data emerged after these 16 interviews. Concerning transferability, other countries and cultures may have other experiences with thirst. The transferability of our results to other health care systems may not be possible as the topic "thirst" is original and not well studied. Another possible limitation of this study was that only a few of the participants were under the age of 40 with a mean age of 52 years. Less experienced physicians might not yet be influenced by the traditional way of perceiving thirst and dry mouth in PC. In this study, there were three different interviewers, which may have influenced the quality of the interviews. In addition,

telephone interviews may have affected the study, as this interview technique will affect the quality of the interview as the body language is missing, that may otherwise give additional information. However, several researchers participated to criticise the analysis and to see other views on the data. These researchers were from different areas such as nursing, medicine, bio-medicine and ethics and this could be seen as an "investigator triangulation" and is, therefore, a strength [32].

## Author Contributions

**Conceptualization:** Maria Friedrichsen, Tiny Jaarsma, Pier Jaarsma, Micha Milovanovic, Marit Karlsson, Anna Milberg, Hans Thulesius, Christel Hedman, Nana Waldréus, Anne Söderlund Schaller.

**Data curation:** Maria Friedrichsen, Tiny Jaarsma, Pier Jaarsma, Helene Ångström, Micha Milovanovic, Marit Karlsson, Anna Milberg, Hans Thulesius, Christel Hedman, Nana Waldréus, Anne Söderlund Schaller.

**Formal analysis:** Maria Friedrichsen, Caroline Lythell, Tiny Jaarsma, Pier Jaarsma, Helene Ångström, Micha Milovanovic, Marit Karlsson, Anna Milberg, Hans Thulesius, Christel Hedman, Nana Waldréus, Anne Söderlund Schaller.

**Funding acquisition:** Maria Friedrichsen, Tiny Jaarsma, Pier Jaarsma, Micha Milovanovic, Marit Karlsson, Anna Milberg, Hans Thulesius, Christel Hedman, Nana Waldréus, Anne Söderlund Schaller.

**Investigation:** Maria Friedrichsen, Tiny Jaarsma, Pier Jaarsma, Helene Ångström, Micha Milovanovic, Marit Karlsson, Anna Milberg, Hans Thulesius, Christel Hedman, Nana Waldréus, Anne Söderlund Schaller.

**Methodology:** Maria Friedrichsen, Caroline Lythell, Tiny Jaarsma, Pier Jaarsma, Helene Ångström, Micha Milovanovic, Marit Karlsson, Anna Milberg, Hans Thulesius, Christel Hedman, Nana Waldréus, Anne Söderlund Schaller.

**Project administration:** Maria Friedrichsen.

**Resources:** Maria Friedrichsen.

**Software:** Maria Friedrichsen.

**Supervision:** Maria Friedrichsen, Tiny Jaarsma.

**Validation:** Maria Friedrichsen, Tiny Jaarsma, Pier Jaarsma, Micha Milovanovic, Marit Karlsson, Anna Milberg, Hans Thulesius, Christel Hedman, Nana Waldréus, Anne Söderlund Schaller.

**Writing – original draft:** Maria Friedrichsen, Caroline Lythell, Tiny Jaarsma, Pier Jaarsma, Helene Ångström, Micha Milovanovic, Marit Karlsson, Anna Milberg, Hans Thulesius, Christel Hedman, Nana Waldréus, Anne Söderlund Schaller.

**Writing – review & editing:** Maria Friedrichsen, Tiny Jaarsma, Pier Jaarsma, Helene Ångström, Micha Milovanovic, Marit Karlsson, Anna Milberg, Hans Thulesius, Christel Hedman, Nana Waldréus, Anne Söderlund Schaller.

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
