## [Decision Letter · Decision Letter 0]

3 Jul 2023

PONE-D-23-01549Thirst or dry mouth, drips or not?

– A qualitative study of palliative care physicians experiences regarding thirst in dying patientsPLOS ONE

Dear Dr. Friedrichsen,

Thank you for submitting your manuscript to PLOS ONE. After careful consideration, we feel that it has merit but does not fully meet PLOS ONE’s publication criteria as it currently stands. Therefore, we invite you to submit a revised version of the manuscript that addresses the points raised during the review process.

We look forward to receiving your revised manuscript.

Kind regards,

Taofiki Ajao Sunmonu

Academic Editor

PLOS ONE

Journal Requirements:

Additional Editor Comments (if provided):

Thank you for a nice work on this qualitative study on dying patients with thirst; an often neglected clinical condition a terminal care Medicne. The authors should try to address the concerns of the eminent reviewers that reviwed this article to improve its quality.

Reviewers' comments:

Reviewer's Responses to Questions

**Comments to the Author**

1. Is the manuscript technically sound, and do the data support the conclusions?

Reviewer #1: Partly

Reviewer #2: Yes

2. Has the statistical analysis been performed appropriately and rigorously? 

Reviewer #1: N/A

Reviewer #2: I Don't Know

3. Have the authors made all data underlying the findings in their manuscript fully available?

Reviewer #1: Yes

Reviewer #2: No

4. Is the manuscript presented in an intelligible fashion and written in standard English?

Reviewer #1: Yes

Reviewer #2: Yes

5. Review Comments to the Author

Reviewer #1: Methodology

1. Please clarify how did you reach these four cities? Were they chosen purposive or randomly? How many palliative care units in each city were involved in recruiting participants?

2. Somewhere you wrote the data were collected between 2019-2022 and another place you mentioned the data were collected in 2020-2022. Please correct this.

3. Why there is a long-time distance to complete interviews (from 2019-2022)? Conducting 16 interviews can be done in shorter time.

4. As the data collection is started before Covid-19 and completed after Cobdi-19, how this may affect the study rigor? How did you solve the issue to reduce the consequence of pandemic?

5. What were the exclusion criteria in this study?

6. How was the process of recruiting participants? Did you just ask the head of the department or a senior physician to encourage physicians participate. Any flyer before the recruitment?

7. Why different interviews with different interview methods (face to face and by telephone) were chosen to conduct 16 interviews? What did you before that to reduce bias and increase credibility of study?

8. How did you calculate sample size? Why 16 participants? How many per each

9. How about data collection tool? How was it developed? How many sections and question shad?

10. Any pilot study to test the transferability or understandability of questions?

11. Was the interview in Sweden? If yes, how did you translated the content after transcribing interviews? Who did this?

12. Did you start analysing after completing interviews or during collecting data? Please clarify this.

13. Did you use any information sheet or consent form before involving participants in the study?

Reviewer #2: It is an unique study on thirst among dying patients in palliative care, which is a rather ignored by the physicians durinc terminal xare. I think this study helps us to differentiate between thirst and dry mouth. However, I have a few observations:

1. Title: I find the title very confusing and distracting. 'Thirst or dry mouth in palliative care patients'- might serve the same purpouse.

2. Discussion: it is rich. But I think the comparison with rodent is not necessary

6. PLOS authors have the option to publish the peer review history of their article (what does this mean?). If published, this will include your full peer review and any attached files.

Reviewer #1: No

Reviewer #2: No

---

## [Author Response · Author response to Decision Letter 0]

31 Jul 2023

Further point by point respoinses 31 of July:

1. Please amend the title either on the online submission form or in your so that they are identical. 

The titles are now identical! 

2. We note your Data Availability statement: "Data cannot be shared publicly because of ethical reasons. Data are available from thefrom the first author on request for researchers who meet the criteria for access to confidential data."Please address the following:

a) At this time disclose the reason for this data restriction and provide a point of contact, preferably an email, for the ethical body mandating the restriction of this data. Please note that this point of contact cannot be an author of this study.Please address the following:

a) At this time disclose the reason for this data restriction and provide a point of contact, preferably an email, for the ethical body mandating the restriction of this data. Please note that this point of contact cannot be an author of this study.

b) If there are no restrictions on this dataset, please provide your data by uploading it as a Supporting Information file or depositing it in a stable repository.

We have now added some sentences about this: Availability of data and materials

As the data for this study is qualitative (interview transcripts), it is sensitive, potentially identifying, and not able to be publicly shared. Requests to access these data can be sent to The Swedish Ethical Review Authority (registrator@etikprovning.se) quoting the study approval number (Ref. 2019–04347).

Point- by- point response to reviewers

Response to editor about journal requirements

Thank you very much for your wise and clear comments! See our answers below! 

We have added a numbered table of contents to our manuscript with different levels of headings, as we believe that it may be easier for readers to distinguish between main themes and sub-themes.

Comment Answer Changed sentence

1When submitting your revision, we need you to address these additional requirements.

 We have considered PLOS ONE's style requirements and used symbol legends for different contributions, author by line and affiliations and corresponding authorship. We have also used different levels of heading in our manuscript according to the journals requirements. See title page and different heading in the whole manuscript.

 The tables are now included in the manuscript. Pages: 7, 9, 10.

 The ethical statement is now placed under the heading “material and methods”, 2.2 sampling and setting. p. 6. Page 6. Ethics committee approval was obtained from The Swedish Ethical Review Authority (Ref. 2019–04347). The study was conducted in accordance with the terms of the Helsinki Declaration, and written informed consent was obtained from each participant.

4. Please review your reference list to ensure that it is complete and correct. If you have cited papers that have been retracted, please include the rationale for doing so in the manuscript text, or remove these references and replace them with relevant current references. Any changes to the reference list should be mentioned in the rebuttal letter that accompanies your revised manuscript. If you need to cite a retracted article, indicate the article’s retracted status in the References list and also include a citation and full reference for the retraction notice. I have checked the reference list and hope that it is okay! 

Additional Editor Comments (if provided):

Thank you for a nice work on this qualitative study on dying patients with thirst; an often neglected clinical condition a terminal care Medicne. The authors should try to address the concerns of the eminent reviewers that reviwed this article to improve its quality. Thank you very much! We are grateful for the comments made! 

Response to reviewer 1. 

Comment Answer Changed sentence

 Thank you very much for your constructive comments to improve our manuscript! We have considered these, and answered each of your questions below. 

Reviewer #1: Methodology

1. Please clarify how did you reach these four cities? Were they chosen purposive or randomly? How many palliative care units in each city were involved in recruiting participants?

2. Somewhere you wrote the data were collected between 2019-2022 and another place you mentioned the data were collected in 2020-2022. Please correct this. Thank you for this question! The cities were chosen purposive, but they had to have advanced palliative units. Thank you for reminding us to clarify this.

PLOS journals require authors to make all data necessary to replicate their study’s findings publicly available without restriction at the time of publication. When specific legal or ethical restrictions prohibit public sharing of a data set, authors must indicate how others may obtain access to the data.

We have added this: 1. Please amend the title either on the online submission form or in your so that they are identical.

2. We note your Data Availability statement: "Data cannot be shared publicly because of ethical reasons. Data are available from thefrom the first author on request for researchers who meet the criteria for access to confidential data."

PLOS journals require authors to make all data necessary to replicate their study’s findings publicly available without restriction at the time of publication. When specific legal or ethical restrictions prohibit public sharing of a data set, authors must indicate how others may obtain access to the data.

b) If there are no restrictions on this dataset, please provide your data by uploading it as a Supporting Information file or depositing it in a stable repository.

Thank you for noticing this! We have now changed to the correct years on page 6.

 Page 5. Data were collected in four purposively selected sized cities in Sweden with populations of between 44,000 to 1,000,000 during 2019-2022. All cities had advanced palliative care units, with physicians specialised in palliative medicine. One palliative care unit was chosen in each city. How many particpants/unit is added in table 1, page 7.

Page 6. Data were collected in 2019-2022 using face-to-face recorded interviews in 2019-2020 (n=10) and telephone interviews in 2021- 2022 (n=6).

3. Why there is a long-time distance to complete interviews (from 2019-2022)? Conducting 16 interviews can be done in shorter time. Yes, you are completely right! This was due to the fact that COVID-19 had just started and that all resources were focused on Covid-19 in Sweden. Several doctors did not have time to participate in the interviews because of this. Therefore, I kept in touch with the medical directors or senior doctors in the meantime to find out when we could start again. We have added a sentence about this on page 6. Page 6. Data were collected in 2019-2022 using face-to-face recorded interviews in 2019-2020 (n=10) and telephone interviews in 2021- 2022 (n=6).

4. As the data collection is started before Covid-19 and completed after Covid-19, how this may affect the study rigor? How did you solve the issue to reduce the consequence of pandemic?

 Good question! Many doctors were tired and exhausted by the COVID-19 pandemic. In addition, this may have affected the study, at least in terms of doing face-to-face interviews. We had to do phone interviews instead to overcome this fact. We have added a sentence about this as a weakness, in the discussion on page 20. See page 20!

In addition, telephone interviews may have affected the study, as this interview technique will affect the quality of the interview as the body language is missing, that may otherwise give additional information. 

5. What were the exclusion criteria in this study? We did not use any exclusion criteria as the inclusion criteria’s helped us to reach our appropriate study group. 

6. How was the process of recruiting participants? Did you just ask the head of the department or a senior physician to encourage physicians participate. Any flyer before the recruitment?

 All medical directors at each unit were asked to participate in the study before the study started, as this is a requirement to receive an ethical approval. Before all interviews started, all physicians received information about the study by e-mail by their medical director or senior physician. The physicians completed a written informed consent. They were assured that all their information would be kept confidential. All participants were informed that participation in the study was voluntary and that they could withdraw their participation at any time without explanation. 

We have added some sentences about this e-mail information and ethics committee approval. Page 5-6. Before all interviews started, all physicians received information about the study by e-mail by their medical director or senior physician. Ethics committee approval was obtained from The Swedish Ethical Review Authority (Ref. 2019–04347). The study was conducted in accordance with the terms of the Helsinki Declaration, and written informed consent was obtained from each participant.

7. Why different interviews with different interview methods (face to face and by telephone) were chosen to conduct 16 interviews? What did you before that to reduce bias and increase credibility of study?

 Due to the covid-19 pandemic, we had to do telephone interviews instead to reach a saturation of fact. We used semi-structured interviews (now added on p 6), which have lower validity than the structured interview, but higher validity that the unstructured interview. However, different interview techniques will affect the quality of the interview as the interviewer only can hear the interviewee, but not see the body language, that may otherwise give additional information. That is a weakness of the study, which we now have added on page 19. We have also written when we did our analysis on page 8.

Bias was reduced by using investigator triangulation. The interviewers did not know about the first analysis so they were not influenced by that. In a qualitative analysis the researcher have to question each step in the analysis, not just looking for confirming data, but for different data to see thirst from as many angles as possible. See page 20 about investigator triangulation. Page 6-7. The semi-structured interview guide was developed by the research team, and addressed physicians’ experiences and views of thirst and ethical challenges (ethical challenges not reported in this study) in patients at the end of life (Table 2). A pilot interview was conducted to test the understanding of the questions and the flow of the questions. No changes were made after the pilot interview.

Page 20. In addition, telephone interviews may have affected the study, as different interview techniques will affect the quality as the body language is missing, that may otherwise give additional information.

Page 8. Two authors (HÅ, MF) independently read all 10 transcripts that were conducted before the covid-19 pandemic. 

In 2021- 2022, when the remaining interviews were conducted the analysis continued. No new data emerged after 16 interviews.

8. How did you calculate sample size? Why 16 participants? How many per each In qualitative studies, it is uncommon to calculate sample size. Qualitative researchers try to reach data saturation. Data saturation is reached when there is enough information and depth in data to replicate the study when the ability to obtain additional new information has been attained, and when further coding is no longer feasible. Saturation is the point at which ‘additional data do not lead to any new emergent themes’. 

I am not sure what you mean by how many per each? Maybe participants/unit? However, we have added this in table 1, on page 7. We think that it will give a better understanding of the participants. Page 8. No new data emerged after 16 interviews.

Geographic location

Palliative care unit 1 6

Palliative care unit2 3 Palliative care unit 3 1

Palliative care unit 4 6

9. How about data collection tool? How was it developed? How many sections and question shad? We used a semistructured interview guide, developed by the research team to address both those practicalities and clinical work as well as the ethical issues around thirst in the end of life. The interview guide had 10 questions and are now available on p. 9. See page 6-7. The semi-structured interview guide was developed by the research team, and addressed physicians’ experiences and views of thirst and ethical challenges (ethical challenges not reported in this study) in patients at the end of life (Table 2).

Page 9. see the interview guide in the table!

10. Any pilot study to test the transferability or understandability of questions? Yes, we did one pilot interview and tested the interview guide for the understanding of our questions and for the flow. However, we also have different professions in our research team who criticised the interview guide for better understanding and logical flow of the interview. What topics should come first? What follows less “naturally”? Avoiding potentially embarrassing questions. We have added this on page 6. Page 6. The semi-structured interview guide was developed by the research team and addressed physicians’ experiences and views of thirst and ethical challenges (ethical challenges not reported in this study) in patients at the end of life (Table 2). A pilot interview was conducted to test the understanding of the questions and the flow of the questions. No changes were made after the pilot interview.

11. Was the interview in Sweden? If yes, how did you translated the content after transcribing interviews? Who did this? Yes, the study was conducted in Sweden. The analysis was done in the Swedish language. When the study had been scrutinised and criticised by our research team, then the study was translated by the first author together with an English native speaker in the research team. It was then sent back to the rest of the research team and after that to a professional transcriber for language editing and understanding. 

12. Did you start analysing after completing interviews or during collecting data? Please clarify this. Good point! As we had to wait for the last interviews, we started to analyse after 10 interviews. Then we made the analysis of the 6 interviews and concluded that we did not need further interviews. We have added this in the paper. Page 8.Two authors (HÅ, MF) independently read all 10 transcripts that were conducted before the covid-19 pandemic. 

In 2021- 2022, when the remaining interviews were conducted the analysis continued. No new data emerged after 16 interviews.

13. Did you use any information sheet or consent form before involving participants in the study?

 Before any interview started, all physicians received information about the study by e-mail by their medical director or senior physician. The physicians completed a written informed consent, some of them just before the interview; the other sent it back by mail or scanned by e-mail. They were assured that all their information would be kept confidential, that they had the right to withdraw the study without any explanation. 

 Page 5-6. Before the interviews started, all physicians received information about the study by e-mail by their medical director or senior physician. 

Ethics committee approval was obtained from The Swedish Ethical Review Authority (Ref. 2019–04347). The study was conducted in accordance with the terms of the Helsinki Declaration, and written informed consent was obtained from each participant.

Response to reviewer 2. Thank you for taking your time to review our manuscript and for the wise comments. 

Comment Answer Changed sentence

Reviewer #2: It is an unique study on thirst among dying patients in palliative care, which is a rather ignored by the physicians durinc terminal xare. I think this study helps us to differentiate between thirst and dry mouth. However, I have a few observations:

 Thank you! 

1. Title: I find the title very confusing and distracting. 'Thirst or dry mouth in palliative care patients'- might serve the same purpouse.

 Thank you! We agree and have now changed the title. Page 1. Thirst or dry mouth in dying patients? 

– A qualitative study of palliative care physicians’ experiences 

2. Discussion: it is rich. But I think the comparison with rodent is not necessary Thank you! Yes, this is perhaps a little bit too ambitious and does not fit in. We have removed the sentence on page 17. Page 17.

Data Availability statement

---

## [Editor Report · Decision Letter 1]

2 Aug 2023

Thirst or dry mouth in dying patients?

– A qualitative study of palliative care physicians’ experiences

PONE-D-23-01549R1

Dear Dr. Friedrichsen,

We’re pleased to inform you that your manuscript has been judged scientifically suitable for publication and will be formally accepted for publication once it meets all outstanding technical requirements.

Kind regards,

Taofiki Ajao Sunmonu

Academic Editor

PLOS ONE

Additional Editor Comments (optional):

Great works. The authors have addressed the concerns of the reviewers of this article. Great works.
---

## [Editor Report · Acceptance letter]

7 Aug 2023

PONE-D-23-01549R1 

Thirst or dry mouth in dying patients? – A qualitative study of palliative care physicians’ experiences 

Dear Dr. Friedrichsen:

I'm pleased to inform you that your manuscript has been deemed suitable for publication in PLOS ONE. Congratulations! Your manuscript is now with our production department. 

Kind regards, 

on behalf of

Dr. Taofiki Ajao Sunmonu 

Academic Editor

PLOS ONE